# The Synergistic Effect of Phosphonic and Carboxyl Acid Groups for Efficient and Stable Perovskite Solar Cells

**DOI:** 10.3390/ma16237306

**Published:** 2023-11-24

**Authors:** Kaihuai Du, Aili Wang, Yue Li, Yibo Xu, Lvzhou Li, Ningyi Yuan, Jianning Ding

**Affiliations:** 1Jiangsu Collaborative Innovation Center for Photovoltaic Science and Engineering, Jiangsu Province Cultivation Base for State Key Laboratory of Photovoltaic Science and Technology, School of Materials Science and Engineering, Changzhou University, Changzhou 213164, Chinadingjn@yzu.edu.cn (J.D.); 2School of Mechanical Engineering, Institute of Technology for Carbon Neutralization, Yangzhou University, Yangzhou 225127, China

**Keywords:** interface passivation, anchoring, charge transfer, self-assembled small molecule, large area

## Abstract

Reducing the interfacial defects between the perovskite/electron transport layer (ETL) is the key point to improving the efficient and stable performance of perovskite solar cells (PSCs). In this study, two self-assembled molecules ((aminomethyl)phosphonic acid and glycine) with different functional groups (phosphonic acid (-H_2_PO_3_) and carboxylic acid (-COOH)) were mixed to form the buried bottom interface of PSCs. The synergistic effect of -H_2_PO_3_ with its higher anchoring ability and -COOH with its fast carrier transport improved the performance of PSCs. Additionally, the SnO_2_ modified by mixed self-assembly molecules (M-SAM) showed a more appropriate energy level alignment, favoring charge transport and minimizing energy loss. In addition, the amine group (-NH_2_) on the two small molecules effectively interacted with uncoordinated Pb^2+^ in perovskite and improved the quality of the perovskite films. Consequently, the (FAPbI_3_)_0.992_(MAPbBr_3_)_0.008_ PSCs with M-SAM reached a PCE of 24.69% (0.08 cm^2^) and the perovskite modules achieved a champion efficiency of 18.57% (12.25 cm^2^ aperture area). Meanwhile, it still maintained more than 91% of its initial PCE after being placed in nitrogen atmosphere at 25 °C for 1500 h, which is better than that of the single-SAM and control devices. Further reference is provided for the future commercialization of perovskite with efficient and stable characteristics.

## 1. Introduction

Planar perovskite solar cells (PSCs) have gradually been considered to be a commercially promising device due to their small current–voltage hysteresis, high power conversion efficiency and simple preparation process [1,2,3]. In the past decade, the power conversion efficiency (PCE) of PSCs has increased from 3.8% to 26.1% [4,5,6]. The electron transport layer (ETL) in the typical n-i-p structure has electron-selective contact with the perovskite absorption layer. Concurrently, the upper surface of the ETL also affects the number of nucleation points and the grain growth of the absorption layer, and ETLs with low-defect concentrations are of decisive significance for improving the performance of PSCs devices [7,8]. The commonly reported ETLs in the n-i-p structure are titanium oxide (TiO_2_) and tin oxide (SnO_2_). However, compared with TiO_2_, the SnO_2_ substrate has the potential for the best ETL due to its high-electron mobility and easy preparation.

The common preparation methods of SnO_2_ when used as electron transport layer mainly include spin-coated SnO_2_ nanocrystals and chemical bath deposition (CBD) of SnO_2_. Different from the spin-coated SnO_2_ method, the SnO_2_ film prepared via CBD is more compact and uniform. Moreover, the CBD method is not limited by the size of substrate and is more beneficial to the future commercial development of PSCs [9]. However, there are hydroxyl (-OH) and oxygen vacancy (OV) defects on the surface of SnO_2_ film prepared via CBD, which significantly increase the non-radiative recombination of the ETL/Perovskite interface, thus changing the performance of PSCs and accelerating their degradation [10]. At the same time, a large number of defects on the ETL/perovskite layer interface are not conducive to the energy level matching of the charge transfer, and lead to serious non-radiative recombination at the interface [11,12]. In order to solve these problems, researchers have made many attempts which can be divided into SnO_2_ body passivation and surface passivation. For body passivation, it has been reported that the addition of phosphonate ethanolamine (PE) to the aqueous colloid of SnO_2_ can significantly reduce the number of Sn dangling bonds in SnO_2_, which is beneficial for improving the electrical properties of SnO_2_. At the same time, the amino group (NH_2_) of PE can interact with the uncoordinated Pb^2+^ in the perovskite active layer, further inhibiting interface defects and improving device performance [13]. However, some researchers have tried to passivate or modify the surface defects of SnO_2_ to reduce the interface defects between SnO_2_ and perovskite [12]. It has been reported that the use of ultraviolet ozone to improve the surface wettability of SnO_2_ while reducing surface defects can achieve a champion efficiency of 17.21%. Some researchers have also reported that the use of KCl aqueous solution to modify the surface of SnO_2_ can passivate the surface defects of SnO_2_ to improve the performance of the device, and finally achieve a champion efficiency of 22.2%. At the same time, using some self-assembled molecules with anchoring groups to passivate the interface is also an effective interface passivation strategy [14]. It has been reported that an anchoring group with strong bonding strength not only shortens the assembly adsorption time and improves the uniformity of adsorption distribution, but also makes the carrier transporting layers (CTL) have a high compatibility with the deposition of the active layer of the perovskite device, thus greatly reducing interface defects [15]. Yang et al. studied a series of self-assembled molecules (SAM) with carboxylic acid groups (-COOH), such as benzoic acid, 3-aminopropionic acid, etc., and found that appropriate interface interactions can significantly reduce the density of defect states and promote charge extraction and transfer at the interface, which increases the efficiency by about 10% compared with solar cells without SAMs, and investigated different chemical interactions between SnO_2_/perovskite [16]. Although small molecules containing -COOH have excellent electron transport capacities due to their special structure, their binding energy on the surface of SnO_2_ is low [17,18,19]. The widely used anchoring group, which is similar to -COOH and has similar anchoring ability, is the phosphonic group (-H_2_PO_3_) [13,20,21]. Its binding energy on the surface of SnO_2_ is much larger than that of the -COOH. Unfortunately, due to the geometric shape and conjugate loss of -H_2_PO_3_, the charge transfer rate of the -H_2_PO_3_ is not competitive with that of -COOH [22]. In addition, it has been reported that the amino group (-NH_2_) contained in the small molecule has the effect of interacting with the uncoordinated Pb^2+^ in the perovskite, which can effectively decrease interface defects [13,23]. Sargent et al. combined ammonium fluoride with CBD-SnO_2_ and found that -NH_2_ could reduce the surface defect sites and match energy levels better with perovskite. This strategy showed a PCE of 23.3% and a higher open circuit voltage (*V*_OC_) [20]. Meanwhile, the application of CBD SnO_2_ to a large area also reflects the problem of non-uniformity. Although CBD SnO_2_ has more commercial potential than spin-coated SnO_2_ nanocrystals, the uniform passivation of SnO_2_ defects is also a problem that must be solved first. Lin et al. developed a simple post-treatment method using periodic acid to modify the SnO_2_ film and increased the proportion of tin oxide (IV) with the help of periodic acid. Consequently, the 3 × 3 cm^2^ perovskite modules with a champion efficiency of 18.10% were shown [24]. Therefore, a strategy that can not only effectively passivate the surface defects of SnO_2_ but also serve as a reference for large-scale commercialization is needed.

In this study, two self-assembled molecules ((aminomethyl)phosphonic acid and glycine) with similar molecular structures but different anchoring groups (phosphonic acid (-H_2_PO_3_) and carboxylic acid (-COOH)) were used as interlayers in SnO_2_/perovskite. Phosphonic acid groups have a stronger anchoring ability on the surface of metal oxides than carboxylic acid groups due to their unique structure, while carboxylic acid groups have their unique characteristics of rapid charge transfer. The different anchoring groups can meet the requirements of the buried bottom interface for carrier transport (-COOH) and anchoring ability (-H_2_PO_3_), so as to solve the defect problem of single functional group. Photovoltaic performance may be further improved by using the complementary advantages of the two groups while improving the anchoring capacity and charge transfer. In addition, the same partial amine group (-NH_2_) on the two small molecules effectively interact with uncoordinated Pb^2+^ in the perovskite layer, which can decrease interface defects. The energy level between SnO_2_ and the perovskite layers was further adjusted by mixed self-assembly molecules (M-SAM). Finally, the PCE of PSC was increased from 22.72% to 24.69% (0.08 cm^2^) and the *V*_OC_ was increased from 1.12 V to 1.16 V. Meanwhile, the stability of the device modified by M-SAM was significantly improved, which maintained more than 91% of its initial PCE after 1500 h and 100 h placed at 25 °C and 65 °C in nitrogen atmosphere. In addition, the 4 × 4 cm^2^ (12.25 cm^2^ aperture area) perovskite modules achieved the best efficiency of 18.57%. This study demonstrates an effective strategy for passivating the surface of SnO_2_, which improves the PCE of PSCs while taking into account stability. This complementary synergistic passivation strategy provides a simple and effective method for future high-performance and durable commercial devices.

## 2. Experimental Section

**Materials.** Hydrochloric acid (HCl, 37 wt.% in water) was purchased from Sinopharm (Shanghai, China). SnCl_2_·2H_2_O (>99.98%) and urea (>99.0%), thioglycolic acid (TGA, ≥99.0%), chlorobenzene (CB, ≥99.9%), isopropanol (IPA, ≥99.9%) and acetonitrile (≥99.9%) were purchased from Sigma-Aldrich (Shanghai, China). Dimethyl sulfoxide (DMSO) (≥99.9%), and dimethylformamide (DMF) (≥99.9%) were purchased from Alfa Aesar (Shanghai China). FAI (≥99.5%), PbI_2_ (99.99%), MAPbBr_3_ (≥99.5%), MACl (≥99.5%), PEAI (≥99.5%), Spiro-OMeTAD (≥99.8%), 4-tert-Butylpyridine (tBP, ≥96%), Co-TFSI salt (≥99.5%) and Li-TFSI salt (≥99.5%) were purchased from Xi’an Polymer Technology Corp (Xi’an, China). (Aminomethyl)phosphonic acid (≥98%) and glycine (99%) were purchased from Adamas (Shanghai China).

**CBD SnO_2_.** The SnO_2_ layer was deposited on the surface of FTO by chemical bath deposition (CBD). The CBD solution was prepared by mixing 5.625 g of urea, 1.2375 g of SnCl_2_·2H_2_O, 5.625 mL of HCl, 112.5 μL of TGA, and 450 mL of deionized water (DI water). The FTO substrates and the CBD solution were loaded into a glass reaction vessel and reacted at 94 °C in an oil bath for 5.5 h. Then, the SnO_2_-deposited FTO were removed from the reaction vessel and cleaned via sonication with DI water and ethanol for 10 min each.

**Preparation of perovskite solution.** The perovskite precursor solution was prepared by mixing 1.4 M FAI, 1.53 M PbI_2_, 0.5 M MACl, and 0.8 mol% MAPbBr_3_ in 1 mL mixture solvent (volume ratio, DMF:DMSO = 8:1).

**Device fabrication.** The FTO/SnO_2_ substrates were annealed at 170 °C for 60 min and treated with UV-Ozone for 20 min. Then the FTO/SnO_2_ substrate was immersed in a small molecule aqueous solution for 20 min, then dried with nitrogen, and finally placed in a 100 °C hot stage for annealing for 5 min. The prepared FTO FTO/SnO_2_/SAM were transferred to the glove box. The perovskite film was spin-coated on the substrates at 1000 rpm for 10 s, and 5000 rpm for 30 s, and chlorobenzene was dripped at 20 s after starting, then annealed at 100 °C for 60 min. For the 2D perovskite passivation, PEAI (15 mM in IPA) was deposited at 5000 rpm for 30 s. For HTL, the 25 μL Spiro-OMeTAD solution was spin-coated on the perovskite layer at 3800 rpm for 30s. The Spiro-OMeTAD solution was prepared by dissolving 101.92 mg of Spiro-OMeTAD, 24.4 μL Li-TFSI salt (520 mg·L^−1^ in acetonitrile solution), 49.6 μL FK209 salt (300 mg·in 1 mL acetonitrile solution), and 45.4 μL tBP solution in 1 mL chlorobenzene. Finally, 100 nm of Au was deposited via thermal evaporation.

## 3. Results and Discussion

The electrostatic surface potential (ESP) diagram and structural formula of P-SAM and C-SAM are shown in Figure 1a and Appendix A, respectively. The dipole moment value of P-SAM (2.11 D) was higher than that of C-SAM (1.66 D). The dipole moment can reflect the polarity of the molecule, and it is easy to obtain the relative level of the van der Waals force of the two molecules; thus P-SAM was more easily anchored on the surface of SnO_2_ than C-SAM was. Figure 1b shows a flow chart of the preparation of the buried bottom interface. The glass/FTO/SnO_2_ substrate was immersed in a M-SAM aqueous solution with a volume ratio of 1:1 (1 mM in deionized water) for 20 min, and we then used N_2_ to blow away the residual solution on the surface of SnO_2_, and annealed it at 100 °C for 5 min. Meanwhile, this immersion method using SAM has the potential to be applied to a large area. The perovskite films on SnO_2_, SnO_2_/P-SAM, SnO_2_/C-SAM and SnO_2_/M-SAM were characterized by scanning electron microscopy (SEM) and X-ray diffraction (XRD). It is evident from Figure 1c–f and Appendix A that the average grain size of perovskite films was similar, but the grain boundary was improved after M-SAM modification (Appendix A). Meanwhile, this method is also applicable to large areas; Appendix A show M-SAM, as a buried interface, can enhance the uniformity of perovskite grains. The enhanced hydrophilicity of the SnO_2_/SAM films (Appendix A) from -NH_2_ in SAM molecules are helpful for the deposition of perovskite precursor solutions and the growth of better-quality perovskite films, which is consistent with the SEM results [25]. The XRD spectra in Figure 1g show that the perovskite films with an M-SAM interlayer have stronger crystallinity, which indicates that M-SAM regulates crystallinity.

The effect of M-SAM modification on the band structures was further studied by ultraviolet photoelectron spectroscopy (UPS) in Figure 1h,i and ultraviolet-visible spectrophotometer (UV-Vis) in Appendix A. The band-gaps of SnO_2_ and SnO_2_/M-SAM were 3.94 and 3.97 eV, respectively. The Fermi level (−*E*_F_ = *E*_cutoff_ − 21.22 eV) and the conduction band (CB) were both up-shifted with the M-SAM modification and thus coordinate the interface energy level arrangement. The CB of SnO_2_ and SnO_2_/M-SAM were −6.39 eV and −5.52 eV, respectively (the details are showed in Appendix A). In addition, the transmittance of SnO_2_ films was not affected by SAMs modification. Correspondingly, a better energy band alignment was achieved between ETL/perovskite, favoring efficient electron transport (Figure 1j) [26].

The interaction mechanism of P-SAM, C-SAM and M-SAM in the buried bottom interface was further explored (Figure 2). In Figure 2a, the Fourier transform infrared spectroscopy (FTIR) spectra shows the P-O vibration shifts from 1167.88 cm^−1^ in pure P-SAM to 1175.17 cm^−1^ of the SnO_2_/P-SAM and the C-O vibration shifts from 1135.25 cm^−1^ in pure C-SAM to 1129.93 cm^−1^ of the SnO_2_/C-SAM, confirming the interaction of P-SAM and C-SAM with SnO_2_, respectively. Similarly, the interaction between the SAM and perovskites is shown in Figure 2b. It can be clearly seen from the moment of initial N-H stretching peak moves from 3162.40 (P-SAM) to 3107.96 cm^−1^ (PbI_2_/P-SAM) and from 3177.48 (C-SAM) to 3166.48 cm^−1^ (PbI_2_/C-SAM), respectively, confirming that both small molecules interact with PbI_2_ [23]. The X-ray photoelectron spectroscopy (XPS) measurements was further studied and the effect of SAM on surface chemical environment was discussed. The Sn 3d peaks in the pure SnO_2_ at 495.28 and 486.88 eV, corresponding to Sn 3d_3/2_ and Sn 3d_5/2_, respectively, shifted to higher binding energies after SAM modifications in Figure 2c (SnO_2_/P-SAM: 495.40 and 486.98 eV, SnO_2_/C-SAM: 495.38 and 486.97 eV, SnO_2_/M-SAM: 495.44 and 487.02 eV). The results show that both P-SAM and C-SAM had chemical interactions with SnO_2_ [27]. As shown in Figure 2d, two peaks fitted at 532.5 eV and 531.1 eV corresponded to the -OH on the surface of SnO_2_ and the saturated lattice oxygen in the SnO_2_ film. Compared with the pure SnO_2_, the calculated areas of -OH in the modified SnO_2_ are all reduced. This indicates that more chemisorbed off-lattice oxygen is transferred to the lattice oxygen, and that the oxygen atom in SnO_2_ exists as O^2−^. The reduced -OH is beneficial for improving the electron transport and suppressing the nonradiative recombination [28]. Meanwhile, in Figure 2e, two peaks fitted at 143.14 and 138.31 eV of Pb 4f spectrum for pure PbI_2_ film, corresponding to Pb 4f_5/2_ and Pb 4f_7/2_, respectively, shifted towards higher binding energies for all SAM-modified samples (SnO_2_/P-SAM: 143.32 and 138.41 eV; SnO_2_/C-SAM: 143.35 and 138.49 eV; SnO_2_/M-SAM: 143.47 and 138.63 eV). The results indicate that the interaction occurs between P-SAM, C-SAM, M-SAM and perovskite.

To further investigate the effect of SAM layers on the carrier transport dynamics, steady-state photoluminescence (SSPL) and time-resolved photoluminescence (TRPL) tests were performed (Figure 3a–d). From the analysis of perovskite, the SnO_2_/SAM/perovskite displays lower PL signals than that of the SnO_2_/perovskite and the SnO_2_/M-SAM/perovskite possesses the lowest PL intensity among the SAM-modified samples. The TRPL curves in Figure 3b exhibit how the average time constant (τ_ave_) of perovskite film on SnO_2_, SnO_2_/P-SAM, SnO_2_/C-SAM and SnO_2_/M-SAM decreased from 1.482 μs to 1.294 μs, and 1.142 μs and 1.056 μs, respectively [29]. The detailed carrier lifetime is listed in Appendix A. The decreased PL intensity and TRPL lifetime indicates that P-SAM-, C-SAM- and M-SAM-modification inhibits non-radiative recombination due to the passivated interface defects of SnO_2_ and the optimized perovskite grain quality [19]. In general, the fast decay process (τ_1_) is related to the charge extraction and transport, and the slow decay (τ_2_) is related to the radiative recombination of the bulk perovskites. The value of τ_1_ decreased from 0.020 μs (SnO_2_/perovskite) to 0.013 μs (SnO_2_/M-SAM/perovskite), which proves that M-SAM-modification is beneficial for the extraction of electrons at the buried interface. In addition, by comparing the SSPL and TRPL results of perovskite films’ deposition on the glass substrate without ETL (Figure 3c,d), the samples show a significantly opposite trend and the corresponding fitted results are presented in Appendix A. The SSPL intensity and the average lifetime of the perovskite films grown on the modified glass were significantly improved, which is attributed to the passivation of interface defects and the optimized grain quality of perovskite [29,30]. Then, the effect of SAM on perovskite films was further studied by PL intensity imaging (Figure 3e,h and Appendix A). Compared with the control films, a weaker PL emission was observed from the SnO_2_/M-SAM/perovskite, which is consistent with the SSPL spectrum shown in Figure 3a. In addition, the SnO_2_/M-SAM/perovskite film exhibited more a uniform PL emission in the substrate compared to the control. These results further prove that M-SAM-modified SnO_2_ can passivate the defects on the surface of SnO_2_ and reduce the non-radiative recombination at the interface to improve the performance of the device. Additionally, the corresponding carrier lifetimes were summarized in Appendix A.

In short, P-SAM, C-SAM, and M-SAM interlayers in the buried bottom exhibited strong interactions with SnO_2_ and perovskite, and thus suppressed the surface defects and facilitated carrier transport. Therefore, the device structure of FTO/SnO_2_/SAM/perovskite/Spiro-OMeTAD/Au (Appendix A) was constructed to investigate the effect of SAM on photovoltaic performance. Appendix A shows the statistical analysis of the PSCs performance on the devices with different molar ratios of P-SAM and C-SAM. The best PCE of the PSC was achieved at a ratio of 1:1 and the corresponding photocurrent density–voltage (*J–V*) curves under AM 1.5 illumination are presented in Figure 4a. The champion PCE of the device using M-SAM increased from 22.72% (control), 23.99% (with P-SAM), 23.56% (with C-SAM) to 24.69%. Compared with the control PSCs, the increase in PCE was mainly due to the obvious increase in *V*_OC_ (from 1.12 to 1.16 V) and *J*_SC_ (from 24.37 to 25.54 mA/cm^2^). In addition to this, a laser-etched perovskite solar module (PSM) with an area of 4 × 4 cm^2^ and an aperture area of 12.25 cm^2^ were also made, and the related *J–V* curve is exhibited in Figure 4b. The PSMs with M-SAM-modifications achieved an excellent PCE of 18.57%, while the PCE of the control PSMs was only 16.07%. A stabilized photocurrent of 23.77 mA/cm^−2^ and an efficiency of 24.25% were obtained at 1.02 V (Figure 4c). The stability was further evaluated. The PCE of the unpackaged device with an M-SAM modification remained at 91% and 92.4% of the initial PCE after exposure to nitrogen atmosphere at 25 °C for 1500 h (Figure 4d) and at 65 °C for 105 h (Figure 4e), respectively. Meanwhile, the control device retained 82% and 79.2% of its original PCE. The improved stability of the M-SAM-modified device is attributed to the synergistic effect of P-SAM and C-SAM, as well as to energy level regulation and improved film quality. In addition to device stability, the perovskite films under different aging conditions were also tested. Figure 4f shows the XRD patterns of SnO_2_/perovskite and SnO_2_/M-SAM/perovskite films before and after aging at 60 °C and 40% relative humidity (RH). After 100 h of aging, the SnO_2_/M-SAM/perovskite film showed a much weaker additional δ-FAPbI_3_ phase diffraction peak than that of the control sample; the result implies that the presence of M-SAM could effectively suppress the phase transition induced by an exposure to moisture, which is consistent with the results in Appendix A. It is therefore further proved that the M-SAM can improve the defects of the SnO_2_/Perovskite interface, and that a stable perovskite film provides an important guarantee for the stability of the later device.

Figure 5a shows the external quantum efficiency (EQE) spectra of the control and the M-SAM-modified PSC, and that the related integrated currents were 24.21 mA/cm^2^ and 24.47 mA/cm^2^, respectively, which are consistent with the result derived from the *J–V* curves. In addition, the EQE spectra of M-SAM-modified PSC was improved in the ultraviolet region, which is attributed to the passivation of Pb^2+^ via -NH_2_ in M-SAM, which reduces the deep-level defects of perovskite films. The dark *J–V* curves (Figure 5b) show the PSCs with M-SAM modifications had a reduced leakage current, ascribing to a lower trapping density and carrier recombination at the interface of SnO_2_/perovskite through M-SAM modification. A quantitative characterization of the defects was obtained by space charge limited current (SCLC) test with the electron-only device (Figure 5c). The trap filling limit voltage (*V*_TFL_) of the device with the M-SAM modification was significantly reduced. The trap state densities were 2.22 × 10^16^ cm^−3^ and 1.94 × 10^16^ cm^−3^ for the control and M-SAM-modified devices, respectively, indicating that the M-SAM can passivate the defects (the parameters were listed in Appendix A) [30]. The Nyquist impedance in Figure 5d shows that the recombination resistance (*R*_rec_) (2117 Ω) of the device with M-SAM is higher than the control (1201 Ω), suggesting a reduced charge recombination at the ETL/PVK interface by M-SAM. Additionally, M-SAM-modified devices achieved higher extracted the built-in potentials (*V*_bi_, 0.95 V) from Mott–Schottky plots (Figure 5e) compared to the control device (0.92 V). The enlarged *V*_bi_ contributed to the higher *V*_OC_, following the same trend of *V*_OC_ values shown in the *J*–*V* curves [30]. The dependence of *V*_OC_ on light intensity (P_light_) was fitted to obtain the ideality factor n (Figure 5f). The M-SAM-modified PSCs possessed the lower n value (1.58) compared to the control (1.93), indicating that the trap-assisted recombination of PSCs was significantly reduced after M-SAM treatment. It has been proven that using M-SAM as the buried interface has a positive effect on improving device performance.

## 4. Conclusions

Aiming at the passivation problem of surface defects commonly used in SnO_2_ ETL, we developed a synergistic effect strategy by combining the strong anchoring ability of phosphonic acid (-H_2_PO_3_) in metal oxide surfaces and the better charge transfer ability of carboxylic acid (-COOH). At the bottom of the interface, the -H_2_PO_3_ in P-SAM was used to enhance the anchoring ability and the -COOH in C-SAM was used to transport carriers quickly. In addition, the molecules interacted with both SnO_2_ from -H_2_PO_3_ and -COOH with Sn and perovskite from -NH_2_ and Pb^2+^, thus improving the energy level arrangement and quality of perovskite films. The (FAPbI_3_)_0.992_(MAPbBr_3_)_0.008_ PSCs with mixed SAM modification passivated the defect density and accelerated the charge extraction and transport. Consequently, the champion PCE of 24.69% and *V*_OC_ of 1.16 V were achieved. Meanwhile, a PCE of 18.57% was obtained for a 4 × 4 cm^2^ (12.25 cm^2^ aperture area) perovskite module. At the same time, more than 91% of the initial PCE was maintained after storage at 25 °C and 65 °C for 1500 h and 100 h under nitrogen atmosphere. This work improves the device efficiency by mixing two complementary self-assembled small molecules to passivate ETL/perovskite interface defects while taking into account stable improvements. This method can improve the passivation uniformity CBD SnO_2_ surface, and is simple and suitable for future commercialization. The reuse of M-SAM passivation solutions need to be further researched in future works.

## Figures and Tables

**Figure 1 materials-16-07306-f001:**
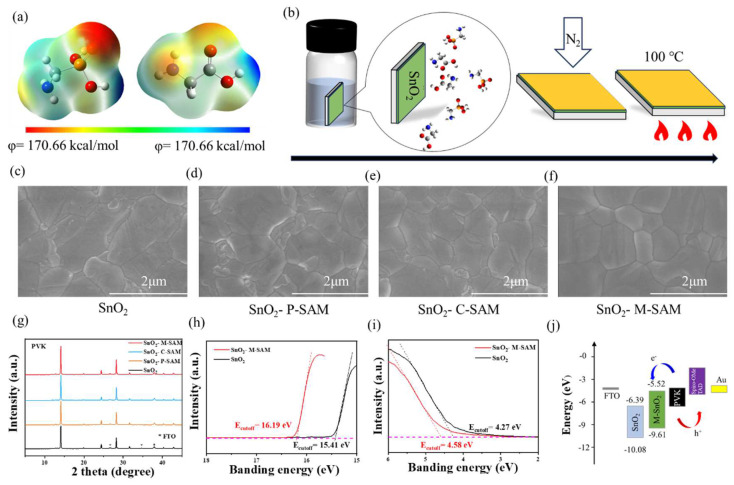
(**a**) ESP map of P-SAM (Left) and C-SAM (Right). (**b**) Diagram of the preparation work of M-SAM. SEM of perovskite films, (**c**) pristine SnO_2_, (**d**) SnO_2_/P-SAM, (**e**) SnO_2_/C-SAM and (**f**) SnO_2_/M-SAM as substrates. (**g**) XRD patterns of perovskite films on SnO_2_, SnO_2_/P-SAM, SnO_2_/C-SAM and SnO_2_/M-SAM. (**h**,**i**) UPS spectra of SnO_2_ and SnO_2_/M-SAM films. (**j**) Energy level diagram of FTO, SnO_2_, SnO_2_/M-SAM, and perovskite.

**Figure 2 materials-16-07306-f002:**
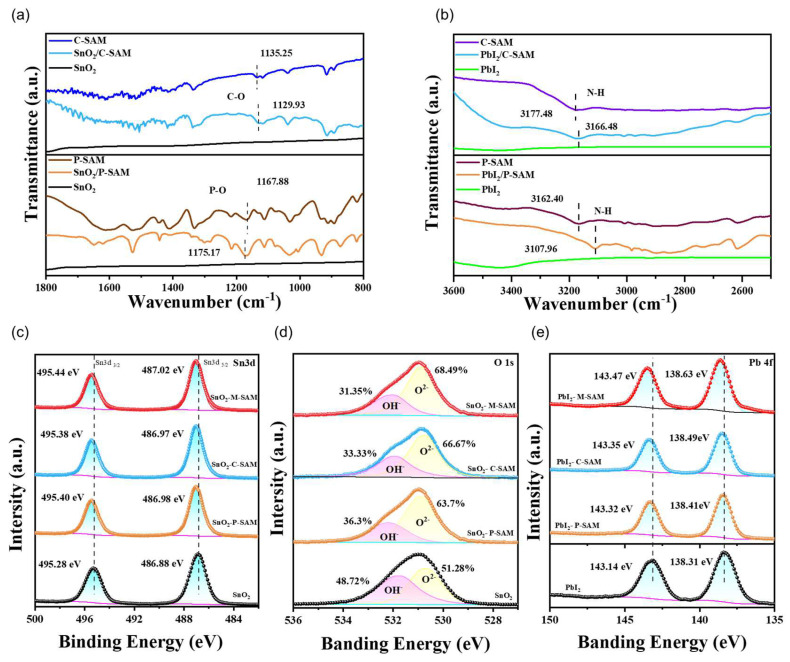
FTIR spectra of (**a**) SnO_2_, SnO_2_/P-SAM, SnO_2_/C-SAM, P-SAM, and C-SAM films, (**b**) PbI_2_, PbI_2_/P-SAM, PbI_2_/C-SAM, P-SAM, and C-SAM films. XPS spectra of (**c**) Sn 3d, and (**d**) O 1s for the pristine SnO_2_, SnO_2_/P-SAM, SnO_2_/C-SAM and SnO_2_/M-SAM films, (**e**) Pb 4f for the PbI_2_, PbI_2_/P-SAM, PbI_2_/C-SAM and PbI_2_/M-SAM films.

**Figure 3 materials-16-07306-f003:**
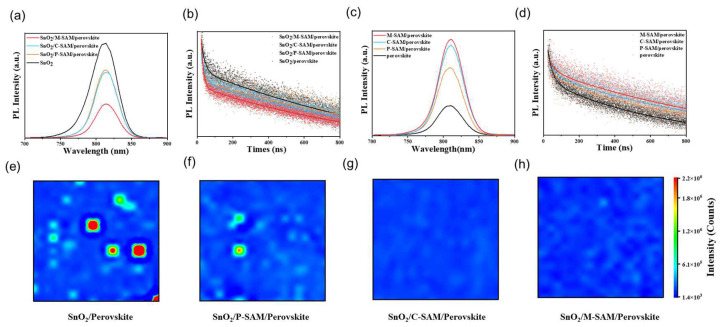
(**a**) SSPL and (**b**) TRPL curves of SnO_2_/perovskite, SnO_2_/P-SAM/perovskite, SnO_2_/C-SAM/perovskite and SnO_2_/M-SAM/perovskite films. (**c**) SSPL and (**d**) TRPL curves of perovskite, P-SAM/perovskite, C-SAM/perovskite and M-SAM/perovskite films. (**e**–**h**) PL intensity imaging of SnO_2_/perovskite, SnO_2_/P-SAM/perovskite, SnO_2_/C-SAM/perovskite and SnO_2_/M-SAM/perovskite films (size of 0.6 × 0.6 mm^2^).

**Figure 4 materials-16-07306-f004:**
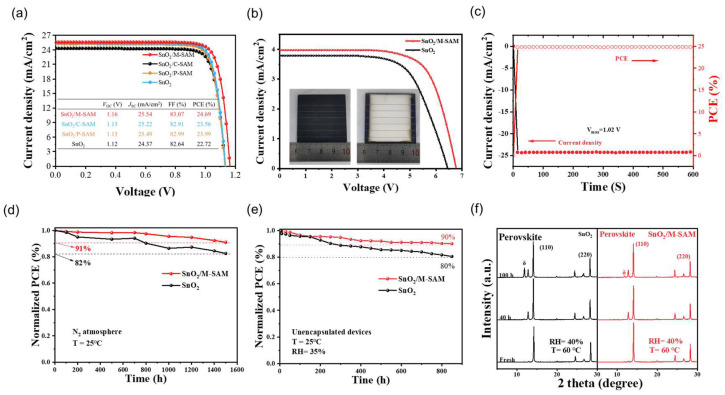
*J*–*V* curves of the champion PSCs for (**a**) 0.08 cm^2^ and (**b**) 4 × 4 cm^2^ modules (**c**) Stable power output curves under maximum power-point conditions. Long-term stability test under nitrogen atmosphere (**d**) at 25 °C and (**e**) at 65 °C. (**f**) XRD patterns for SnO_2_/perovskite and SnO_2_/M-SAM/perovskite films aged at ≈40%RH, ~60 °C.

**Figure 5 materials-16-07306-f005:**
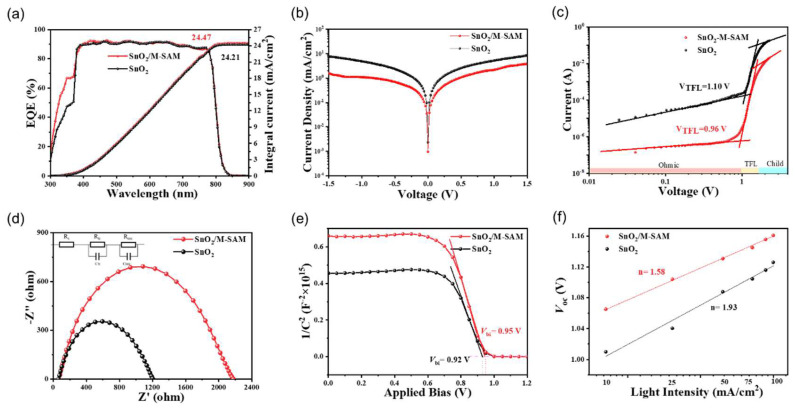
(**a**) The EQE spectra and integrated *J*_SC_ of SnO_2_ and SnO_2_/M-SAM devices, (**b**) dark *J–V* curves of SnO_2_ and SnO_2_/M-SAM devices, (**c**) dark *J–V* curves for the devices with structure of FTO/SnO_2_(M-SAM)/PVK/PCBM/Ag, (**d**) Nyquist plots of SnO_2_ and SnO_2_/M-SAM devices, (**e**) Mott–Schottky plots of SnO_2_ and SnO_2_/M-SAM devices, (**f**) dependence of *V*_OC_ and light intensity dependence of SnO_2_ and SnO_2_/M-SAM devices.

## Data Availability

The research data are contained within the article.

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
