# Peer review of "The Synergistic Effect of Phosphonic and Carboxyl Acid Groups for Efficient and Stable Perovskite Solar Cells"

_materials, 2023, doi:10.3390/ma16237306_

Round 1

Reviewer 1 Report

Comments and Suggestions for Authors

The article "The synergistic effect of phosphonic and carboxyl acid groups 2 for efficient and stable perovskite solar cells" shows merit in the field of research and can be processed after revision.

1) By using the current method, how the energy losses be reduced and improve the efficiency? An analytical explanation is needed.

2) Provide comparative analysis for the current design with Single-SAM regarding the controlling of devices

3) A comparative analysis with literature in the form of a table will provide more insight of work and present the novelty.

4) Application-oriented presentation in conclusion will be appreciated.  

Comments on the Quality of English Language

NA

Author Response

Journal: Materials

Manuscript ID: materials-2731118

Title of Paper: “The synergistic effect of phosphonic and carboxyl acid groups for efficient and stable perovskite solar cells”

Authors: Kaihuai Du, Aili Wang, Yue Li, Yibo Xu, Lvzhou Li, Ningyi Yuan, Jianning Ding

Thank you for your letter and for the reviewers’ comments concerning our manuscript entitled “The synergistic effect of phosphonic and carboxyl acid groups for efficient and stable perovskite solar cells” (materials-2731118). Those comments are valuable and greatly helpful for revising and improving our paper, as well as the important guiding significance to our researches. We have studied the comments carefully and made corresponding corrections according to the suggestions. Simultaneously, all changes to the manuscript are marked in yellow. We believe the quality of the revision has been further improved on the basis of the reviewers’ comments and hope that the revised manuscript can be acceptable for publishing on your prestigious Materials.

The responses to the reviewer comments are shown as follows:

Comment 1: By using the current method, how the energy losses be reduced and improve the efficiency? An analytical explanation is needed.

Author reply: Thank you for your valuable suggestion. According to your precious suggestions, the analytical explanation of the current method in terms of energy loss is as follows:

(1) For one thing, mixing two small molecules of (aminomethyl)phosphonic acid and glycine as a buried bottom interface can passivate the surface defects of SnO2 (such as hydroxyl groups), which can reduce the non-radiative recombination of the SnO2/perovskite interface and reduce energy loss.

(2) For another, by modifying SnO2 with M-SAM, the energy level state of SnO2 can be improved, so that there is a better energy level arrangement between SnO2 and perovskite, which can better carry out charge transfer and reduce energy loss.

Comment 2: Provide comparative analysis for the current design with Single-SAM regarding the controlling of devices.

Author reply: Thanks for your kind reminding. We feel sorry that we did not provide enough information about the comparative analysis between the two samples. Based on your advice, the comparative analysis of Single-SAM and M-SAM: (aminomethyl)phosphonic acid and glycine have some of the same functional groups and their own characteristic functional groups. Phosphonic acid groups have stronger anchoring ability on the surface of metal oxides than carboxylic acid groups due to their unique structure, while carboxylic acid groups have their unique characteristics of rapid charge transfer. By mixing the two small molecules, the complementary advantages of the two small molecules can be realized, and the stability and charge transfer ability of the buried bottom layer can be improved. And Table 2 shows the J-V parameters of single SAM, M-SAM and control.

Table 2. the J-V parameters of single SAM, M-SAM and the control.

VOC (V)

JSC (mA/cm2)

FF (%)

PCE (%)

Control

1.12

24.37

82.64

22.72

P-SAM

1.13

25.49

82.99

23.99

C-SAM

1.13

25.22

82.91

23.56

M-SAM

1.16

25.54

83.07

24.69

Comment 3: A comparative analysis with literature in the form of a table will provide more insight of work and present the novelty.

Author reply: Thanks for your valuable comment. According to your gracious suggestion, we have added this content in the Supporting information and marked it in blue. And the specific table is as follows:

Table 3. The modification of CBD SnO2 for getting the improved performance of PSCs are listed below.

Surface passivation

VOC (V)

JSC (mA/cm2)

FF (%)

PCE (%)

Mercaptosuccinic acid[1]

1.14

22.95

80.9

21.16

NaBF4[2]

1.10

23.51

80.48

20.82

Periodic acid[3]

1.09

25.02

81.55

22.25

This Work

1.16

25.54

83.07

24.69

[1] Zhang, J. B.; Bai, C.; Dong, Y.; Shen, W. J.; Zhang, Q., Batch Chemical Bath Deposition of Large-area SnO2 Film with Mercaptosuccinic Acid Decoration for Homogenized and Efficient Perovskite Solar Cells. Chemical Engineering Journal 2021, 425, 8.

[2] Soe, K. T.; Thansamai, S.; Thongprong, N.; Ruengsrisang, W.; Muhammad, I. A., Simultaneous Surface Modification and Defect Passivation on Tin Oxide-Perovskite Interfaces using Pseudohalide Salt of Sodium Tetrafluoroborate. Solar Rrl 2023, 7, (1), 15.

[3] Wu, Z. Y.; Su, J. Z.; Chai, N. Y.; Cheng, S. Y.; Wang, X. Y., Periodic Acid Modification of Chemical-Bath Deposited SnO2 Electron Transport Layers for Perovskite Solar Cells and Mini Modules. Advanced Science 2023, 10, (20), 9.

Comment 4: Application-oriented presentation in conclusion will be appreciated.

Author reply: Thank you for your valuable suggestion. According to your suggestion, we modify the conclusion as follows: This work improves the device efficiency by mixing two complementary self-assembled small molecules to passivate ETL/perovskite interface defects while taking into account the stable improvement. This method can improve the passivation uniformity of large-area CBD SnO2 surface, and the method is simple and suitable for future commercialization. Then, the reuse of M-SAM passivation solution needs further research in the later work.

Reviewer 2 Report

Comments and Suggestions for Authors

Reducing the interfacial defects between the perovskite/electron transport layer (ETL) is the key point to get the efficient and stable performance of perovskite solar cells (PSCs). In this study, two self-assembled molecules ((aminomethyl)phosphonic acid and glycine) with different functional groups (phosphonic acid (-H2PO3) and carboxylic acid (-COOH)) were used as the buried bottom interface of PSCs which demonstrated rather high PCE. The paper could be published after revision.

-In this study, two self-assembled molecules ((aminomethyl)phosphonic acid and glycine) with similar molecular structures but different anchoring groups (phosphonic acid (-H2PO3) and carboxylic acid (-COOH)) were used as interlayer in SnO2/perovskite. It should be clearly explained why these molecules were tested as there is big amount of similar materials.

-Chemical structures of the mentioned molecules should be demonstrated.

-The glass/FTO/SnO2 substrate was immersed in a M-SAM aqueous solution with volume ratio of 1:1 (1 mM in deionized water) for 20 min, then the surface was dried with N2 and annealed at 100 °C for 5 min. Why the annealing is useful here ?

- PCEs of the PSCs and other characteristics should be better compared with those described in literature.

-Advantages and disadvantages of  PSCs presented here should be described in comparison with already published similar devices.

Author Response

Authors’ Response to the Reviewer Comments

Journal: Materials

Manuscript ID: materials-2731118

Title of Paper: “The synergistic effect of phosphonic and carboxyl acid groups for efficient and stable perovskite solar cells”

Authors: Kaihuai Du, Aili Wang, Yue Li, Yibo Xu, Lvzhou Li, Ningyi Yuan, Jianning Ding

Thank you for your letter and for the reviewers’ comments concerning our manuscript entitled “The synergistic effect of phosphonic and carboxyl acid groups for efficient and stable perovskite solar cells” (materials-2731118). Those comments are valuable and greatly helpful for revising and improving our paper, as well as the important guiding significance to our researches. We have studied the comments carefully and made corresponding corrections according to the suggestions. Simultaneously, all changes to the manuscript are marked in yellow. We believe the quality of the revision has been further improved on the basis of the reviewers’ comments and hope that the revised manuscript can be acceptable for publishing on your prestigious Materials.

The responses to the reviewer comments are shown as follows:

Comment 1: It should be clearly explained why these molecules were tested as there is big amount of similar materials.

Author reply: Thank you for your valuable suggestion. The reasons why we choose (aminomethyl)phosphonic acid and glycine are as follows:

  1. It has been reported before that the length of the self-assembled small molecule carbon chain will affect the effect of modification, so we choose a shorter carbon chain length.
  2. At present, the phosphonic acid group in the commonly used SAM is the majority of the anchoring group. In order to make up for the shortcomings of its poor electron transfer ability, we selected the carboxylic acid group that also has the anchoring ability to make up for its shortcomings.

iii. Both of them can be formulated into an aqueous solution, which can be mixed in any proportion, taking into account the requirements of future commercial large-scale and green environmental protection.

Comment 2: Chemical structures of the mentioned molecules should be demonstrated.

Author reply: Thank you for your valuable suggestion. (aminomethyl)phosphonic acid (≥98%) and glycine (99%) were purchased form Adamas. Figs 2a and 2b are the IR1 spectra of (aminomethyl) phosphonic acid and glycine, respectively.

Fig 2a The IR1 spectra of (aminomethyl) phosphonic acid.1

Fig 2b The IR1 spectra of glycine.2

[1] The IR1 spectra of (aminomethyl) phosphonic acid. https://www.chemicalbook.com/Spectrum_1066-51-9_IR1.htm

[2] The IR1 spectra of glycine. https://www.chemicalbook.com/Spectrum_56-40-6_IR1.htm

Comment 3: The glass/FTO/SnO2 substrate was immersed in a M-SAM aqueous solution with volume ratio of 1:1 (1 mM in deionized water) for 20 min, then the surface was dried with N2 and annealed at 100 °C for 5 min. Why the annealing is useful here?

Author reply: Thanks for your kind reminding. We 're very sorry we did not explain it clearly in the manuscript. The purpose of the nitrogen drying process is to remove the solution remaining on the surface after FTO/SnO2 immersion, and the heating step is to further dry the wet film.

Comment 4: PCEs of the PSCs and other characteristics should be better compared with those described in literature.

Author reply: Thank you for your valuable comment. We add a table comparing with the published literature.

Table 4. The modification of CBD SnO2 for getting the improved performance of PSCs are listed below.

Surface passivation

VOC (V)

JSC (mA/cm2)

FF (%)

PCE (%)

Mercaptosuccinic acid[1]

1.14

22.95

80.9

21.16

NaBF4[2]

1.10

23.51

80.48

20.82

Periodic acid[3]

1.09

25.02

81.55

22.25

This Work

1.16

25.54

83.07

24.69

[1] Zhang, J. B.; Bai, C.; Dong, Y.; Shen, W. J.; Zhang, Q., Batch Chemical Bath Deposition of Large-area SnO2 Film with Mercaptosuccinic Acid Decoration for Homogenized and Efficient Perovskite Solar Cells . Chemical Engineering Journal 2021, 425, 8.

[2] Soe, K. T.; Thansamai, S.; Thongprong, N.; Ruengsrisang, W.; Muhammad, I. A., Simultaneous Surface Modification and Defect Passivation on Tin Oxide-Perovskite Interfaces using Pseudohalide Salt of Sodium Tetrafluoroborate. Solar Rrl 2023, 7, (1), 15.

[3] Wu, Z. Y.; Su, J. Z.; Chai, N. Y.; Cheng, S. Y.; Wang, X. Y., Periodic Acid Modification of Chemical-Bath Deposited SnO2 Electron Transport Layers for Perovskite Solar Cells and Mini Modules. Advanced Science 2023, 10, (20), 9.

Comment 5: Advantages and disadvantages of PSCs presented here should be described in comparison with already published similar devices.

Author reply: Thank you for your valuable suggestion. Compared with similar devices published, the advantages and disadvantages of this method are as follows:

Advantages: i) it can improve the uniformity of surface passivation of CBD SnO2 in large area; ii) this method is simple and suitable for commercialization;

Disadvantages: The utilization rate of M-SAM solution needs to be further improved.

Round 2

Reviewer 2 Report

Comments and Suggestions for Authors

Accept in present form